# Role of Point-of-Care Gastric Ultrasound in Advancing Perioperative Fasting Guidelines

**DOI:** 10.3390/diagnostics14212366

**Published:** 2024-10-23

**Authors:** Alina Razak, Silva Baburyan, Esther Lee, Ana Costa, Sergio D. Bergese

**Affiliations:** 1Department of Anesthesiology, Stony Brook University Health Science Center, Stony Brook, NY 11794, USA; alina.razak@stonybrookmedicine.edu (A.R.); ana.costa@stonybrookmedicine.edu (A.C.); 2Renaissance School of Medicine, Stony Brook University, Stony Brook, NY 11794, USA; silva.baburyan@stonybrookmedicine.edu (S.B.); esther.lee@stonybrookmedicine.edu (E.L.)

**Keywords:** POCUS, gastric ultrasound, point-of-care ultrasound, GLP1, fasting, perioperative

## Abstract

Pulmonary aspiration in the perioperative period carries the risk of significant morbidity and mortality. As such, guidelines have been developed with the hopes of minimizing this risk by recommending fasting from solids and liquids over a specified amount of time. Point-of-care ultrasound has altered the landscape of perioperative medicine; specifically, gastric ultrasound plays a pivotal role in perioperative assessment. Further, the advent of glucagon-like-peptide-1 receptor agonists, the widespread use of cannabis, and Enhanced Recovery program carbohydrate beverage presents new challenges when attempting to standardize fasting guidelines. This review synthesizes the literature surrounding perioperative fasting guidelines specifically with regard to the use of point-of-care ultrasound in assessing for gastric contents and minimizing the risk of aspiration.

## 1. Introduction

The risk of pulmonary aspiration in the perioperative period remains a concern for anesthesiologists as it has historically been associated with increased morbidity and mortality [1,2]. The American Society of Anesthesiologists (ASA) Closed Claims Project noted that 5% of claims related to pulmonary aspiration resulted in death or permanent severe injury from 2000 to 2014 [3]. It is imperative for anesthesiologists to implement precautions to prevent further incidents of pulmonary aspiration. The 2017 American Society of Anesthesiologists Practice Guidelines for Preoperative Fasting provided updated guidance regarding the nil per os (NPO) status prior to the induction of anesthesia [4]. Via a systematic review, the guidelines detail a preoperative assessment to determine comorbidities and risk of pulmonary aspiration as well as specific fasting requirements for clear liquid, breast milk, infant formula, and solids. Several international societies of anesthesiologists have provided guidelines for NPO status perioperatively and the utility of gastric ultrasound in this setting (Table 1) [4,5,6,7,8,9,10,11].

Since then, evolving patient landscapes have brought several new issues to the forefront when managing patients in the perioperative period. Point-of-care ultrasound (POCUS), specifically gastric ultrasound, has become a readily available tool for anesthesiologists and may guide perioperative decision making [12]. Concurrently, the increasing use of glucagon-like-peptide-1 (GLP) receptor agonists, approved by the Food and Drug Administration for the treatment of type 2 diabetes and obesity, has prompted concerns regarding delayed gastric emptying [13]. The legalization of recreational cannabis since 2012 has raised concerns for perioperative management given the risk of hyperemesis and gastroparesis [14]. Enhanced Recovery After Surgery (ERAS) protocols encouraging carbohydrate-containing beverages preoperatively may confound traditional fasting guidelines [15]. This is a sampling of the complexities when determining the risk of pulmonary aspiration and developing an anesthetic management plan for these patients.

As patient comorbidities and the advent of new technology alter the landscape of perioperative management, new concerns have arisen. This review seeks to provide an overview of the present literature regarding perioperative fasting guidelines specifically in relation to the utility of gastric ultrasound in the perioperative setting and for patients utilizing GLP-1 agonists, cannabis, and Enhanced Recovery program carbohydrate beverages.

## 2. Use of Point-of-Care Gastric Ultrasound for Assessing Aspiration Risk

Point-of-care ultrasound (POCUS) has become increasingly accessible for anesthesiologists for utilization during the perioperative period. Specifically, gastric ultrasound has garnered interest by physicians in an attempt to further assess delayed gastric emptying and minimize the risk of pulmonary aspiration.

### 2.1. Indications for Perioperative Gastric Ultrasound

Gastric ultrasound can be used preoperatively in conjunction with a thorough history of NPO status when examining patients. When the NPO status is uncertain, such as for patients with an altered mental status, conflicting histories from both the patient and caretaker, or pre-existing patient comorbidities associated with delayed gastric emptying (such as diabetic gastroparesis), gastric ultrasound can serve as an objective measure of quantifying full stomach risk [16]. It is hypothesized that healthy patients who have adequately fasted for an elective procedure have a low pre-test probability of having an ultrasound with increased gastric volume and may not warrant a POCUS exam. However, further research is necessary to further elucidate this [12]. Caution should be utilized in patients with altered gastric anatomy as most risk calculations have been validated on patients with typical anatomy.

### 2.2. Overview of Steps to Performing Gastric Ultrasound

Once an appropriate patient has been identified, a standardized approach to performing gastric ultrasound can be followed, first with the patient in a supine position and then in the right lateral decubitus position (Figure 1). A curvilinear low-frequency probe is frequently used for adult patients, while a high-frequency probe may be more appropriate for pediatric patients [12]. The antrum has a reliable location, and, particularly in the right lateral decubitus position, it also has a more dependent location for gastric contents to gather while air is displaced laterally away from the sonographic beam [17]. The transducer is optimally placed in the midline under the xiphoid; at this location, the antrum is identified inferior to the left hepatic lobe and anterior to the retroperitoneal pancreas (Figure 2). An optimal view includes the abdominal aorta and superior mesenteric vessels which are distal to the antrum [18]. When empty, the antrum has been described as an ovoid shape with the anterior and posterior stomach walls in close proximity to one another; however, after ingestion of fluid, the antrum becomes distended with hypoechoic content, while after ingestion of solid food, the antrum may demonstrate a “frosted glass” appearance as artifacts from the interface between mucosa and air preclude the sonography of posterior structures [12,17,18].

A measurement of the cross-sectional area of the gastric antrum in the right lateral decubitus position can be inputted into a standardized model created by Perlas et al., which utilizes the cross-sectional area and age to determine the volume of gastric fluid present [19]. Low risk has been quantified as either an empty antrum or a gastric volume less than or equal to 1.5 mL/kg (Grade 1 antrum), while a suggestion of high risk occurs with a gastric volume greater than 1.5 mL/kg (Grade 2 antrum) or a presence of solid material in the antrum [20].

### 2.3. Validity of Gastric Ultrasound Measurements

The validation of gastric ultrasound is undertaken in healthy volunteers to determine the utility of the imaging modality. It is critical to determine the training process for anesthesiologists who have not undertaken gastric ultrasound imaging previously, as well as determine the reproducibility of results among providers. Given the presence of a learning curve when beginning POCUS scanning, a cohort study was conducted by Arzola et al. [21]. Anesthesiologists underwent a classroom learning portion prior to an interactive learning session with hands-on practice. The cohort of learners then underwent an assessment scanning healthy volunteers with a pre-determined prandial status, and it was determined that a mean of 33 cases was necessary for the six anesthesiologists to reach a 95% success rate for determining a qualitative NPO status (identifying an antrum that is empty, contains liquid, or contains solids). In a similar study of two regional anesthesiologists adept with peripheral ultrasonography, the providers determined qualitative results on gastric ultrasound with 96% accuracy; however, determining the quantitative cross-sectional area to determine gastric volume may require a more formal training process [22]. In a prospective study, healthy volunteers consumed a set volume of apple juice prior to undergoing gastric ultrasound by three independent and experienced gastric ultrasound sonographers resulting in accurate and reproducible results with a 2.7% median difference amongst antrum volume measurements, suggesting reproducible results amongst experienced providers [23].

### 2.4. Utility of Gastric Ultrasound in Perioperative Assessment

Quantitative studies have been performed in the perioperative period to determine the correlation between gastric volume as measured by ultrasound versus nasogastric tube output. A cohort of 50 patients selected for elective surgery, both with diabetes and without diabetes and who had fasted for eight hours, underwent preoperative quantitative gastric ultrasound assessment followed by the nasogastric tube aspiration of stomach contents once intubated under general anesthesia [24]. Overall, the patients with diabetes had larger gastric volume both on ultrasound as calculated by antral cross-sectional area and through physical fluid measurement from the nasogastric tube. The two volumes were largely correlated, with a correlation coefficient of 0.951 in the right lateral decubitus position. Volume correlation was also studied by Gultekin et al. [25]. The group studied healthy patients who first underwent upper endoscopy with a complete aspiration of stomach contents. Afterwards, they consumed varying volumes of a biscuit with fruit juice and underwent gastric ultrasound examination by skilled practitioners. For up to a volume of 200 mL, a strong linear correlation between ingested volume and cross-sectional area was demonstrated.

For patients with diabetes, a meta-analysis of 18 observational studies found that most studies indicated a trend towards larger gastric volume on gastric ultrasound compared to patients without diabetes. However, there may be confounding factors such as glycated hemoglobin that may be accounting for the effect resulting in the need for further analysis [26].

When utilized for preoperative assessments, gastric ultrasound may lead to alterations in the anesthetic management of patients undergoing elective and emergency surgeries. In a prospective observational study, patients at an increased risk of pulmonary aspiration (for example, those with active nausea and vomiting, intestinal obstruction, non-fasted patients, or reflux) were enrolled and taken to the operating room. The anesthesiologist overseeing the patient’s care was asked to rate the aspiration risk based on clinical judgment from 1 to 10 both before and after gastric ultrasound sonography and determine if any changes were necessary from the proposed management plan. The authors found that approximately 4% of all patients required a more conservative management plan (for example, endotracheal intubation versus supraglottic airway device), while approximately 15% of patients could have been managed more liberally (for example, avoiding endotracheal intubation if risk of aspiration appeared reduced following gastric ultrasound) [27]. A similar trend was noted by Alakkad et al., where patients scheduled for elective surgery who did not comply with fasting guidelines underwent gastric ultrasound [28]. The anesthesiologist responsible for the care of the patient made an initial pre-test plan for anesthesia management (proceed with the case, delay the case, or cancel the case) and was then informed of the results and determined a post-test plan for management. Management changes occurred in 71% of patients in both directions; some patients who were initially delayed proceeded on time, while patients who were initially scheduled to proceed were delayed given residual findings on gastric ultrasound. Gastric ultrasound is best utilized in conjunction with clinical decision making to determine the need for a more conservative or liberal management strategy.

## 3. Role of Perioperative Gastric Ultrasound for Patients Utilizing GLP-1 Agonists

Glucagon-like-peptide-1 (GLP-1) receptor agonists have gained increasing popularity for their efficacy in managing type 2 diabetes and obesity. By promoting glucose-dependent insulin secretion, decreasing glucagon release, and delaying gastric emptying, GLP-1 agonists modulate postprandial glucose levels and enhance feelings of satiety [29] Of particular importance to perioperative medicine is the effect of GLP-1 agonists on gastric emptying. While a direct causal relationship between GLP-1 agonists and aspiration events has not been established yet, several studies have suggested an increased risk of perioperative aspiration events in patients taking these medications [30]. As such, there is a growing need to understand the extent to which GLP-1 agonists interrupt gastric motility to provide appropriate recommendations for patients undergoing anesthesia.

### 3.1. Mechanism of Delayed Gastric Emptying with GLP-1 Agonists

Although the overall concept that GLP-1 agonists delay gastric emptying has been established in the literature, the specific mechanisms and timeline of GLP-1-induced gastroparesis seem to be multifactorial. In fact, short-acting and long-acting GLP-1 agonists have shown to yield different results in gastric emptying. The gastric emptying rate was significantly delayed by lixisenatide, a short-acting agonist, while long-acting agonists such as liraglutide and dulaglutide had limited effects on gastric emptying [31]. Furthermore, some studies suggest that the prolonged stimulation of GLP-1 receptors through the chronic use of GLP-1 agonists may induce tachyphylaxis and have a diminished effect on gastric emptying [32,33].

One study showed that prolonged (24 h), intermittent (two 4.5 h infusions separated by 20 h), and acute (4.5 h) injections of GLP-1 all increase gastric retention. However, this effect was attenuated in participants that were subjected to prolonged GLP-1 stimulation [34]. This finding is further supported by two studies, both of which found no statistically significant difference in gastric emptying compared to placebo after 12 and 20 weeks of semaglutide use [35,36].

The effect of GLP-1 agonists on gastric emptying may also vary with postprandial time. For example, liraglutide significantly slowed gastric emptying during the first postprandial hour compared to control, but not at five hours after meals [37]. Van Can et al. also found a significant slowing of gastric emptying at one hour after meals with five weeks of liraglutide treatment compared to placebo, but not at five hours. In addition, they found that a 3.0 mg, liraglutide treatment group experienced a 23% reduction in one-hour gastric emptying compared to control (*p* = 0.007), while a 1.8 mg liraglutide treatment group only experienced a 13% reduction in one-hour gastric emptying (*p* = 0.14) [38]. This is consistent with previous studies that suggest that GLP-1 agonists may suppress gastric emptying in a dose-dependent manner [39,40]. Overall, the gastroparetic effects of GLP-1 agonists seem to almost completely resolve within a few hours of administration. While eight weeks of daily liraglutide use resulted in a substantial delay (25–40%) in gastric emptying, gastric motility was found to return to baseline four hours after meals [41].

To date, most clinical studies focus on GLP-1 agonist-induced gastric emptying delay in the context of esophagogastroduodenoscopy (EGD), which helps visualize the stomach and quantify the residual gastric content. A matched pair case–control study of 205 pairs with diabetes scheduled to undergo EGDs showed that GLP-1 agonist treatment group had higher proportions of gastric residue compared to the control group (5.4% and 0.49%, respectively, *p* = 0.004) [42]. Similarly, a single-center retrospective chart review found a greater residual gastric content of 24.2% (*p* < 0.001) in patients who took semaglutide despite sufficient preoperative fasting, compared to a 5.1% increase in residual gastric content in patients who had not taken semaglutide prior to elective EGDs [43]. Interestingly, gastrointestinal symptoms like nausea and vomiting were also associated with increased residual gastric content in those with perioperative semaglutide use (prevalence ratio = 16.5 [95% CI 9.08–34.91]), suggesting the predictive value of these symptoms in assessing aspiration risks [43]. A meta-analysis by Nascimento et al. in 2024 demonstrated that patients utilizing GLP-1 agonists were significantly more likely to have symptoms such as dyspepsia, nausea, and bloating, as well as increased gastric residual volume compared to control when fasting times were equivalent [44].

### 3.2. Investigations of Gastric Ultrasound for Patients Utilizing GLP-1 Agonists

Investigations are ongoing regarding the impact of perioperative gastric ultrasound for patients utilizing GLP-1 agonists (Table 2). Sen et al. performed a cross-sectional study of patients presenting on day of elective surgery who receive once-weekly injections of GLP-1 agonists [45]. Three separate blinded anesthesiologists trained in gastric ultrasonography acquired images, performed calculations, and resolved any discrepancies. Compared to the control and despite equivalent NPO status, patients in the GLP-1 agonist group had increased gastric residual contents (56% versus 19%, prevalence ratio = 2.9 [95% CI 1.6–5.0]). The study was limited in their small sample size and inability to measure aspiration events as the results of the gastric ultrasound were disclosed to the anesthesiology team to mitigate any patient harm. Further studies are warranted to provide an in-depth characterization of the time course of gastric emptying following administration of GLP-1 agonists as well as the impact of potential confounding variables (for example, glycated hemoglobin levels or the presence of symptoms concerning for delayed gastric emptying).

A case report by Giron-Arango and Perlas demonstrated a point-of-care use of gastric ultrasound for a patient taking a GLP-1 agonist that may be implemented by practitioners on the day of surgery [46]. Gastric ultrasound can be utilized in the preoperative setting for risk assessment and in shared decision making with the patient regarding the safety of proceeding with the surgery and selection of anesthetic technique.

### 3.3. Recommendations for Perioperative GLP-1 Use

The consensus-based guideline released by the ASA recommends that, in the context of elective procedures, patients on daily GLP-1 agonist dosing hold their medication on the day of the procedure, while those on weekly dosing should hold their medication a week prior to their procedure. However, these recommendations are limited in that they are primarily based on case reports and therefore lack substantial evidence to comment on the optimal duration of fasting for patients on GLP-1 agonists [30].

Although data specifying the exact time of fasting to ensure safety are lacking, the half-life of most commercially available GLP-1 agonists suggests that the current guidelines may not be sufficient. Commonly used long-acting GLP-1 agonists, typically administered once a week, have long half-lives. For example, semaglutide has a half-life of seven days and the long-acting release formulation of exenatide has a half-life of two weeks [47,48]. Since it takes about four to five half-lives for a drug to almost completely be eliminated, current recommendations to hold long-acting GLP-1 agonists for one week may not be sufficient to prevent delayed gastric emptying and its downstream effects.

There is also some evidence that stopping GLP-1 agonists for a prolonged amount of time may unnecessarily increase the risk of adverse events. Many studies have raised concerns that stopping GLP-1 agonists preoperatively can result in hyperglycemia and related adverse events after surgery, such as poor wound healing and increased length of stay [49]. This highlights the need for more detailed guidelines and alternative medication options for optimal glycemic control while patients withhold their GLP-1 agonists. In addition, several medications are associated with an increased risk of slowing gastric motility, including anticholinergics, narcotic analgesics, antidepressants, calcium channel blockers, and tetrahydrocannabinol. As these medications may impact the effects of gastric slowing even further, a careful review of all the medications and a multidisciplinary approach to medication management is needed for the optimal perioperative management of patients on GLP-1 agonists [50].

Even though the literature shows the effects of GLP-1 agonists on slowing gastric emptying, the perioperative management of GLP-1 agonists is nuanced, and several factors must be considered. Based on available data, it is reasonable to recommend that patients with short-term use of GLP-1 agonists be managed with extra caution [51]. After 12 or more weeks of prolonged use, however, standard fasting times are likely to decrease the chance of increased gastric retention and related aspiration risks. The half-life of GLP-1 agonists must also be carefully considered, as those taking weekly injections may benefit from a longer cessation time than one week, which is the current recommendation. While the effects of GLP-1 agonists are generally expected to be negligible within a few hours postprandially, patients undergoing urgent procedures should be managed with increased caution and be treated as patients with a “full stomach” as recommended by the ASA [30]. As a cautionary measure, guidelines from many Societies of Anesthesiologists, particularly those from The United States, Canada, and Australia/New Zealand, suggest POCUS to evaluate gastric volume for patients utilizing GLP-1 agonists as part of a cohesive approach to minimizing aspiration risk (Table 1).

Additional research is needed to further elucidate the optimal perioperative management of patients on GLP-1 agonists based on drug type, dose, and surgical procedure. Further, detailed recommendations regarding patients’ glycemic control and co-prescribed medications are needed. While the risk of aspiration during induction of general anesthesia remains low, further studies are needed to quantify the increase in gastric contents related to GLP-1 agonist use to encapsulate a holistic perioperative aspiration risk assessment.

## 4. Role of Perioperative Gastric Ultrasound for Patients Utilizing Cannabis

Cannabis is the most commonly used recreational psychoactive substance in the United States and globally. Approximately 192 million people, or 3.9% of the world population aged 15–64 years, used cannabis in 2018 [52,53]. According to the Substance Abuse and Mental Health Services Administration, 22% (61.9 million) of the United States population used cannabis in 2022 [54]. Given its widespread use, it is critical for anesthesiologists to properly screen all patients for cannabis use, understand its effects on physiology, and have the tools necessary to provide safe perioperative care to all patients who use it.

The cannabis plant contains more than 500 compounds, including over 100 cannabinoids [55]. The two main endogenous cannabinoids derived from cannabis are cannabidiol (CBD) and Δ9-tetrahydrocannabinol (THC). CBD, THC, and other synthetic cannabinoids (nabilone, dronabinol) act on cannabinoid receptors to exert their effects. These include cannabinoid receptor 1 (CB1) and cannabinoid receptor 2 (CB2), both of which are inhibitory G-protein-coupled receptors that are distributed widely in the peripheral and central nervous systems [56]. THC is a partial agonist at both CB1 and CB2 receptors, while CBD may act as an inverse agonist or antagonist at these receptors, highlighting the variable effects of cannabinoids. Activation of cannabinoid receptors leads to inhibition of adenylate cyclase activity and decreased intracellular cyclic AMP, leading to the activation of voltage-gated potassium channels and inhibition of calcium channels, thus inhibiting neurotransmitter release [57]. These changes can produce significant alterations in physiology and place cannabis users at a greater risk of certain perioperative complications.

### 4.1. Complications Associated with Cannabis Use and Delayed Gastric Emptying

Some complications of cannabis use in relation to anesthesia are well known. The cannabinoids undergo extensive hepatic first-pass metabolism and are metabolized predominantly by the cytochrome P450 family of isoenzymes (CYP3A4 and CYP2C19), leading to multiple cannabinoid drug interactions [57]. Additionally, cannabis has multiple effects on the cardiovascular system via THC’s stimulation of the sympathetic nervous system while inhibiting the parasympathetic nervous system [58]. A 2020 retrospective cohort analysis demonstrated that surgical patients with an active cannabis use disorder had significantly increased rates of postoperative myocardial infarction compared to those without cannabis use disorder [59].

Cannabis use has also been associated with gastrointestinal effects, most notably delayed gastric emptying, cyclical vomiting, and cannabis hyperemesis syndrome [57]. Both murine and human studies have demonstrated that cannabinoids, THC specifically, inhibit gastrointestinal motility and slow gastric emptying [60,61]. This may put patients at an increased risk of pulmonary aspiration, a rare but life-threatening complication, during the perioperative period. Cannabis has also been reported to be an important, yet lesser-known, cause of gastroparesis. Case reports as well as a double-blind, randomized controlled study demonstrated THC’s ability to slow gastric emptying of solid food from an average of 30 min to 120 min in human subjects [62,63].

A case report was published in 2021 of an otherwise healthy 24-year-old man who used cannabis daily and had presented for a lower extremity open reduction and internal fixation. While he had fasted for longer than eight hours, after induction of general anesthesia and laryngeal mask airway (LMA) placement, 150 mL of gastric contents had filled the LMA. After suctioning and securing the airway with an endotracheal tube, an additional 500 mL of clear gastric contents were suctioned from the stomach. Additionally, the patient had been completely asymptomatic and endorsed no signs of gastric fullness or hyperemesis syndrome [64]. While the connection between THC and delayed gastric emptying is evident, this case raises several important questions, such as the need for updated fasting guidelines for patients who use cannabis as well as protocols for aspiration risk prevention via premedication or gastric ultrasound. Further, the quantity, frequency, and form of cannabis consumption may impact perioperative management.

Additional research is necessary in this area to provide more comprehensive recommendations. As cannabis remains classified as a Schedule I substance by the United States Drug Enforcement Agency, there are multiple barriers to cannabis research efforts [14].

### 4.2. Recommendations for Perioperative Cannabis Use and Role of Gastric Ultrasound

With the growing prevalence of cannabis use and shifting societal attitudes, ongoing research efforts are underway, leading to the gradual release of updated guidelines. In 2023, the first guidelines by an anesthesiology society in the United States regarding cannabis use in the perioperative setting were released by the American Society of Regional Anesthesia and Pain Medicine [57]. Evidence-based recommendations were made based on an extensive literature review and the experience of 13 experts, including anesthesiologists, chronic pain physicians, and a patient advocate. When discussing special perioperative considerations for patients who use cannabis, the group acknowledged that more evidence is needed on the topic of THC use on gastric emptying and the risk of aspiration. Given the concern for possible delayed gastric emptying and the prevalence of cyclical vomiting, patients consuming cannabis may be candidates for preoperative gastric ultrasonography. Recruitment for clinical investigations is ongoing to better elucidate the potential utility and validity of gastric ultrasound in this setting. For now, the recommendation is to continue screening all surgical and procedural patients for cannabis use and to be aware of its effects on physiology, especially in the perioperative period.

## 5. Role of Perioperative Gastric Ultrasound for Patients Utilizing Carbohydrate Drinks as Part of Enhanced Recovery After Surgery Protocols

Enhanced Recovery After Surgery (ERAS) protocols are evidence-based, multimodal, multidisciplinary perioperative approaches to facilitate recovery after surgical procedures [65]. Anesthesia management is vital in reducing postoperative complications and achieving earlier recovery [66]. An exaggerated stress response may complicate any major surgery or procedure and lead to increased pain, paralytic ileus, increased cardiac demand, and respiratory issues. This may cause delayed recovery and increased hospital stays. One of the main goals of the ERAS protocols is to dampen this stress response by successfully implementing preoperative, intraoperative, and postoperative measures aimed at addressing each source of stress [67].

Preoperative patient preparation and assessment is one of the essential components of the ERAS. This includes managing comorbidities like diabetes, hypertension, and chronic obstructive pulmonary disease, as well as lifestyle changes like smoking cessation leading up to surgery [66]. Traditionally, the American Society of Anesthesiologists Practice Guidelines for Preoperative Fasting recommended patients to be NPO for at least eight hours to reduce the risk of pulmonary aspiration [4]. In the most recent 2023 update to the guidelines, patients are recommended to continue drinking clear carbohydrate-containing liquids until two hours preoperatively for procedures requiring general anesthesia, regional anesthesia, or procedural sedation [15]. Carbohydrate loading has been shown to expedite the return of bowel function, reduce insulin resistance, maintain muscle strength, and reduce patient discomfort, anxiety, and hunger [68,69,70]. Additionally, in 31 randomized controlled trials, no pulmonary aspiration was reported between the fasting group and the group consuming carbohydrate-containing clear liquids 2 h preoperatively [15].

### 5.1. Carbohydrate Drinks for Specific Surgical Procedures

Several medical, anesthesia, and surgery factors may put individuals at an increased risk of pulmonary aspiration [71]. In these patients, it is crucial to evaluate fasting guidelines on a case-by-case basis. It has been proposed that patients undergoing upper abdominal surgery, such as bariatric surgery, may be considered at higher risk of aspiration as surgical manipulation may push gastric contents cephalad [72]. Surgeries involving the lithotomy position may contribute to the regurgitation of gastric contents. In addition, laparoscopic procedures may incur a higher risk of aspiration due to pneumoperitoneum, increased intra-abdominal pressure, and the Trendelenburg position. Laparoscopic cholecystectomies (LCs) may be at an even higher risk of aspiration as secretion of gastric acid is increased, and patients may regurgitate or vomit bile-containing fluid [71,73].

#### 5.1.1. Laparoscopic Cholecystectomy

As LCs have become the standard surgical management of cholelithiasis, many institutions have implemented the ERAS protocols to help reduce postoperative complications. A study published in 2021 looked at whether shortening the preoperative fasting period as part of an ERAS protocol in patients undergoing LCs would increase the incidence of intraoperative aspiration. A total of 179 healthy patients undergoing elective LCs were randomized to either the study group, in which they would fast for six hours, with two hours of water deprivation leading up to the surgery, or to the control group, which would fast for 12 h with six hours of water deprivation leading up to surgery. There were no differences in baseline characteristics between groups. The researchers found that the study group had a significantly shorter hospital stay (4.68 ± 1.03 v 6.52 ± 1.22, *p* < 0.001) without increased intraoperative reflux or aspiration. Additionally, patients in the study group reported decreased thirst (*p* < 0.001), decreased hunger (*p* < 0.001), and increased comfort (*p* < 0.001) [74].

Yuan et al. conducted a double-blind randomized controlled trial of healthy adult patients undergoing LCs who were randomized to receive either 300 mL of water or enzyme-hydrolyzed rice flour between two and three hours preoperatively. When assessed with gastric ultrasound by a skilled provider blinded to study group, there was no significant difference in gastric volume between patients consuming enzyme-hydrolyzed rice flour carbohydrate beverage and those consuming water two hours prior to LCs at the time of induction of anesthesia. Further, they found that patients who had oral enzyme-hydrolyzed rice flour solution, a water-soluble carbohydrate, compared to water two to three hours before an LC were more comfortable (*p* < 0.05) and had improved postoperative gastrointestinal function (*p* < 0.05) without incidences of aspiration [75].

#### 5.1.2. Bariatric Surgery

The ERAS protocols have also been successfully adopted in bariatric surgery with various interventions to reduce postoperative complications [76]. Additionally, the use of multimodal stress-minimizing approaches has been shown to shorten functional recovery as well as the length of stay in the hospital [77]. Suh et al. performed a randomized controlled study in 2021, which included 134 patients undergoing either a minimally invasive Roux-en-Y gastric bypass or a sleeve gastrectomy. Patients either remained NPO after midnight prior to surgery (standard group) or consumed two carbohydrate drinks: one the night prior to surgery and another three hours preoperatively (intervention group). There were no episodes of aspiration among the intervention and standard groups. Further, there was no statistically significant difference in the length of hospital stay between groups [78]. These preliminary results suggest that preoperative carbohydrate loading may be safe in bariatric surgery patients.

### 5.2. Carbohydrate Drinks, Aspiration Risk, and Gastric Ultrasound 

The ERAS protocols have been linked to decreased postoperative complications and faster recovery. Carbohydrate loading has specifically been associated with improved patient outcomes, including quicker return to normal bowel function, reduced insulin resistance, and less patient discomfort, anxiety, and hunger [79,80]. Moreover, studies have shown the benefits of carbohydrate loading in patients undergoing bariatric surgery and LCs, surgeries believed to have an increased risk of pulmonary aspiration, up to 2–3 h before the procedure. Although many of these studies are limited by their single-institution design and relatively small sample sizes, especially considering that aspiration is a rare event, the current literature suggests that carbohydrate loading may be safe in these types of surgeries without increased significant risks.

Studies implementing gastric ultrasound have demonstrated the safety of consuming carbohydrate-containing beverages 2 h prior to surgery (Table 3). Cho et al. recruited healthy female participants undergoing a routine elective laparoscopic gynecologic procedure and randomized to either NPO after midnight or NPO after midnight with the exception of a carbohydrate beverage up to 2 h prior to surgery [81]. Skilled sonographers blinded to a study group acquired images and performed cross-sectional-area calculations. There was no significant difference in gastric cross-sectional area in the right lateral decubitus position between groups. Similarly, Chen et al. demonstrated no significant difference in gastric ultrasound results for elderly patients undergoing orthopedic surgery who either consumed a carbohydrate beverage up to 2 h preoperatively or not [82]. While the utility of carbohydrate beverages may be individualized to patient and procedure, gastric ultrasound as a modality can also be utilized to ensure adequate gastric emptying prior to proceeding with the procedure.

## 6. Conclusions

In an effort to minimize the pulmonary aspiration of gastric contents, preoperative fasting guidelines recommend withholding solids for 6–8 h and clear fluids for 2 h prior to anesthesia. Recent studies have shown that reducing fasting times does not increase aspiration risk and leads to reduced postoperative nausea and vomiting and better patient well-being, especially in children [83,84,85,86,87].

Pulmonary aspiration remains a concern for anesthesiologists in the perioperative period. In conjunction with a thorough review of the patient’s history, prescribed and recreational substance use, comorbidities, current symptoms, and fasting status, an anesthetic plan can be formulated with necessary precautions in place. While traditional fasting guidelines remain as a mainstay for anesthesiologists, new patient factors continue to play a role in perioperative management. The use of gastric ultrasound may facilitate perioperative management decision making. Concerns regarding an increased risk of aspiration have been noted with the use of GLP-1 agonists, cannabis, and carbohydrate-containing beverages. Given the complexities of each comorbidity, future studies should quantify precautions taken to avoid aspiration risk and perhaps make further recommendations for prolonged fasting in specific subsets of patients or suggest additional evaluation using gastric ultrasound.

## Figures and Tables

**Figure 1 diagnostics-14-02366-f001:**
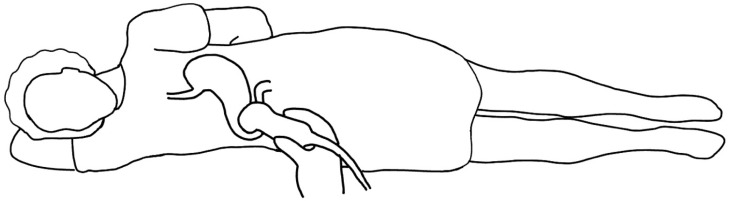
Patient in the right lateral decubitus position for gastric ultrasonography. Drawing designed by authors.

**Figure 2 diagnostics-14-02366-f002:**
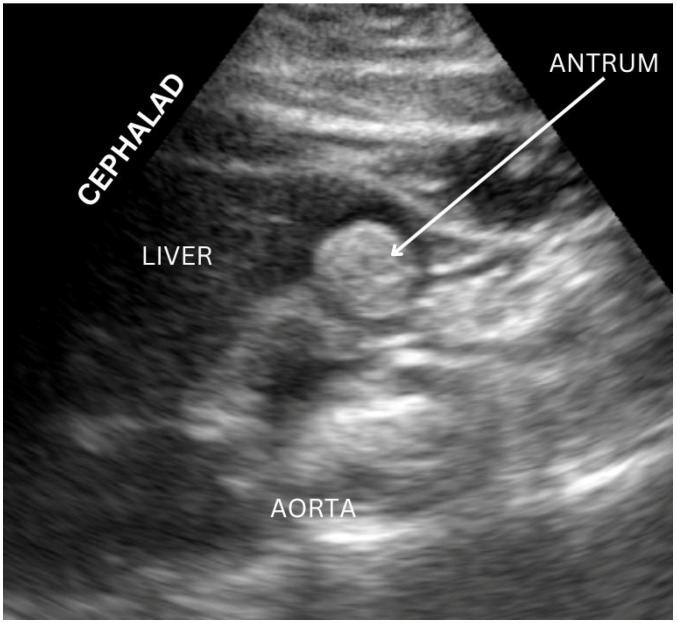
Example of gastric ultrasound image acquisition with gastric antrum visualized. Image originally acquired by authors.

**Table 1 diagnostics-14-02366-t001:** Comparison of international fasting guidelines.

Minimum Fasting Period (Hours)						
	America [4]	Canada [5,6]	Australia and New Zealand [5,7]	Europe [8,9]	Brazil [10]	India [11]
Clear liquids	2	2	2	2	2	2
Breast milk	4	4	3	3–4	4	
Nonhuman milk, including infant formula	6	6	6 (4 h for infants > 12 months)	4–6	6	4
Light meal (toast, clear liquids)	6	6	6	6	6	6
Regular or heavy meal (fried or fatty, meat)	8	8	No comment	6	8	10
Encourage clear fluid intake	Yes	Yes	Yes	Yes	No comment	No
Use of carbohydrate-rich beverages	2	No comment	No comment	2	No comment	No comment
Pharmacological intervention	No routine use	No comment	Consider	No evidence	No comment	No evidence
Gum chewing	No indication to cancel	No comment	No indication to cancel	No indication to cancel	No comment	No indication to cancel
Utility of perioperative gastric ultrasound	Suggested	Suggested	Suggested	Suggested	Suggested	Suggested

**Table 2 diagnostics-14-02366-t002:** Overview of investigations of gastric ultrasound for patients utilizing GLP-1 agonists as part of Enhanced Recovery After Surgery protocols.

Ultrasound for Patients Utilizing GLP-1 Agonists	
Sen et al. [45]	Patients in the GLP-1 agonist group had increased gastric residual contents despite NPO status when assessed via gastric ultrasound
Giron-Arango et al. [46]	Case report of point-of-care gastric ultrasound use for a patient utilizing a GLP-1 agonist

**Table 3 diagnostics-14-02366-t003:** Overview of investigations of gastric ultrasound for patients utilizing carbohydrate drinks as part of Enhanced Recovery After Surgery protocols.

Ultrasound for Patients Utilizing Carbohydrate Drinks as Part of Enhanced Recovery After Surgery Protocols	
Yuan et al. [75]	No significant difference in gastric volume between patients consuming enzyme-hydrolyzed rice flour carbohydrate beverage and those consuming water two hours prior to surgery
Cho et al. [81]	No significant difference in gastric cross-sectional area between NPO after midnight and NPO after midnight with exception of a carbohydrate beverage up to 2 h prior to surgery for patients undergoing gynecological surgery
Chen et al. [82]	No significant difference in gastric ultrasound results between NPO after midnight and NPO after midnight with exception of carbohydrate beverage up to 2 h prior to surgery for elderly patients undergoing orthopedic surgery

## Data Availability

No new data were created or analyzed in this study. Data sharing is not applicable to this article.

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
