# Peer review of "Role of Point-of-Care Gastric Ultrasound in Advancing Perioperative Fasting Guidelines"

_diagnostics, 2024, doi:10.3390/diagnostics14212366_

Round 1

Reviewer 1 Report

Comments and Suggestions for Authors

This review addresses an important topic with is risk for perioperative aspiration and new challenges around it after novel medicines and changes in substance use, and the potential benefit of gastric ultrasound in minimizing this risk.

The manuscript is well presented, the topics covered are broad and the literature reviewed is relevant. I only suggest to the authors two aspects that could help improve the understanding of the manuscript by readers and update its content.

1. Overview of Steps to Performing Gastric Ultrasound. The authors' description is comprehensive but could benefit from adding some images or diagrams illustrating the ultrasound findings and measurements they describe in their text.

2. GLP-1 Agonists and Delayed Gastric Emptying

Recently, a systematic review of clinical trials has provided robust evidence on the impact of glucagon-like peptide-1 receptor agonists in several aspects of the patients undergoing anesthesia or sedation (PMID: 39039540). Authors should consider referencing the results of this work in their review.

Author Response

We thank the reviewer for their thoughtful feedback. Please see attached document for the author's reply to the reviewer.

Reviewer 2 Report

Comments and Suggestions for Authors

This is a well written review. However, I suggest making the following modifications before the official publication.

1. Add some ultrasound images related to gastric ultrasound in Part 2 of the main text.

2. Main headings for sections 3, 4, and 5 should be revised to role of perioperative gastric ultrasound for patients utilizing GLP-1 Agonists, cannabis, carbohydrate drinks, respectively. The corresponding content order also needs to be adjusted.

Author Response

We thank the reviewer for their thoughtful feedback. Please see the attached document for the author's reply to the reviewer.

Reviewer 3 Report

Comments and Suggestions for Authors

A review paper for POC ultrasound for preoperative fasting guidelines is provided. However, previous studies for each section provide too small information. Authors must analyze more previous studies because authors offer more details about agonists except for other information. Authors need to investigate more case reports for perioperative cannabis use. In addition, the authors need to provide several Tables to summarize the characteristics of each section. Therefore, authors must re-write the whole manuscript with enough time. In addition, Figure 1 should obtain copyright permission. Therefore, I must not recommend this submitted manuscript. Except for those critical comments, there are some suggestive comments below.

1. Line spacing between author lists and abstracts needs to be corrected. Authors do not use proper document format.

2. On page 3, space needs to be corrected.

3. Reference format needs to be corrected according to MDPI author guidelines.

4. The Declaration of Interest and Disclosures section needs to be corrected.

5. For laparoscopic cholecystectomy, authors must provide some protocol definitions before provding the surgery data.

6. Authors do not provide how to categorize each part for OC ultrasound for preoperative fasting guidelines such as PubMed.

7. Abbreviations need to be moved before the Reference section.

8. In the manuscript, the reference format should be [ ] which is not ( ). Please check MDPI author guidelines more carefully.

Comments on the Quality of English Language

None

Author Response

(The authors gave the same response as above.)

Round 2

Reviewer 3 Report

Comments and Suggestions for Authors

The authors properly answered the questions and updated them so I can recommend the submitted manuscript can be accepted as it is.